# Parallax attention stereo matching network based on the improved group-wise correlation stereo network

**Xuefei Yu, Jinan Gu** [ORCID]**\*, Zedong Huang, Zhijie Zhang**

School of Mechanical Engineering, Jiangsu University, Zhenjiang 212000, China

\* gujinan@tsinghua.org.cn

**Data Availability Statement:** Three open source datasets are used in our study. Those are free to obtain through the following links: ① Kitti2012 dataset, download link: http://www.cvlibs.net/datasets/kitti/eval_stereo_flow.php?benchmark=

## Abstract

Recent stereo matching methods, especially end-to-end deep stereo matching networks, have achieved remarkable performance in the fields of autonomous driving and depth sensing. However, state-of-the-art stereo algorithms, even with the deep neural network framework, still have difficulties at finding correct correspondences in near-range regions and object edge cues. To reinforce the precision of disparity prediction, in the present study, we propose a parallax attention stereo matching algorithm based on the improved group-wise correlation stereo network to learn the disparity content from a stereo correspondence, and it supports end-to-end predictions of both disparity map and edge map. Particular, we advocate for a parallax attention module in three-dimensional (disparity, height and width) level, which structure ensures high-precision estimation by improving feature expression in near-range regions. This is critical for computer vision tasks and can be utilized in several existing models to enhance their performance. Moreover, in order to making full use of the edge information learned by two-dimensional feature extraction network, we propose a novel edge detection branch and multi-featured integration cost volume. It is demonstrated that based on our model, edge detection project is conducive to improve the accuracy of disparity estimation. Our method achieves better results than previous works on both Scene Flow and KITTI datasets.

## Introduction

The binocular stereo matching task is an imperative, but difficult scientific problem, which aims at computing disparity data for every pixel from a stereo correspondence. Efficient and correct stereo matching methods are necessary for computer vision tasks such as robotic pose estimation and autonomous driving [1,2].

Traditional stereo matching methods usually consist of four steps: initial matching cost calculation, matching cost aggregation, disparity prediction, and post-processing. These can be categorized into global and local algorithms [3]. Local strategies solely use constant measurement windows or changeable windows to calculate the preliminary cost. Global strategies normally treat an optimization task by minimizing a word goal characteristic that incorporates

stereo ② Kitti2015 dataset, download link: http://www.cvlibs.net/datasets/kitti/eval_scene_flow.php?benchmark=stereo ③ Scene flow dataset, download link: https://lmb.informatik.uni-freiburg.de/resources/datasets/SceneFlowDatasets.en.html.

**Funding:** This research was funded by the National Natural Science Foundation of China (No. 51875266). J.G. is the author who received the award, and whose role in our study is to provide experimental equipment and supervise the rationality of our research. you can find this funding at the link of https://isisn.nsfc.gov.cn/egrantindex/funcindex/prjsearch-list.

**Competing interests:** The authors have declared that no competing interests exist.

statistics and smoothness terms. However, traditional algorithms need to manually design feature description operators and cost aggregation strategies, which is not suitable for real-time applications. Complicated hand-craft production steps limit their improvement.

Learning-based stereo-matching methods achieve accurate matching of the corresponding points in the left and right feature maps through exploring feature representations and aggregation algorithms. Those algorithms commonly consist of the following four steps: unary characteristic extraction [4], constructing cost volume [5], value aggregation [6], and disparity prediction [7]. Although the performance on several benchmarks is significantly promoted, drawbacks still remain: Firstly, the predicted edge cues of the disparity map is not accurate enough. Secondly, adopting the strategy of global attention, which is insensitive to the detailed texture regions, resulting in inaccurate disparity estimation in vital areas.

In recent years, researchers are encouraged by the mechanism of human attention and attempt to design some network attention architectures with a CNN to enhance the performance of feature extraction. However, drawbacks still remain: Firstly, these works focus on parallax information in two-dimensional(2D) domain, 2D feature is difficult to fully reflect the three-dimensional(3D) real scenes, while ignoring the more important 3D information. Secondly, due to limited learning capability of a single network structure, the disparity map predicted is not fine enough in near-range regions.

The edge cues of images are the most easily recognized feature by human eyes, in other words, humans can easily find the stereo correspondence by using edge cues of binocular images. Based on this observation, some researches have made partial progress in predicting image edge cues as a single task. In recent years, researchers are encouraged by edge detection (ED) task and use it for disparity estimation project. However, these methods regard disparity prediction and edge detection as a multi-task learning project. Yet, features learned in such multi-task pipelines cannot be fully exploited, which poses a great need for an effective fusion mechanism.

In the context of autonomous driving, the relatively closer area provides larger parallax information, leading to greater risks. To address this problem, more attention should be assigned to this kind of region in the disparity estimation model. In this paper, we propose a high-quality and efficient module for stereo matching and our method achieves better performance on SceneFlow [8] and KITTI [9,10] than previous methods. Specifically, we examine the very important issue of structure design, ***attention***. The importance of attention has been researched particularly in previous methods [11,12]. In our module, the left characteristic map $f_l$ and the corresponding right feature map $f_r$ are packed in the shape of a 3D feature map $f$, which is sent to the parallax attention (PA) stereo module to learn 'what is' to attend $f$. As shown in Fig 1, our structure efficiently improves the accuracy of disparity prediction by improving feature expression in near-range regions. Meanwhile, a novel edge detection branch and a multi-featured integration cost volume are proposed in our network to learn finer texture features, which are vital in the optimization of unary feature extraction tasks. In order to complete the end-to-end disparity prediction task, we assign different weights to edge detection loss and disparity smooth loss. It is demonstrated that achieving high-precision edge feature map is conducive to improve the accuracy of disparity estimation.

Our main contributions can be summarized as follows:

1. We propose a PA module to further improve the accuracy of disparity prediction;

2. An edge detection branch and a multi-featured integration cost volume are proposed in our network architecture to obtain finer texture features;

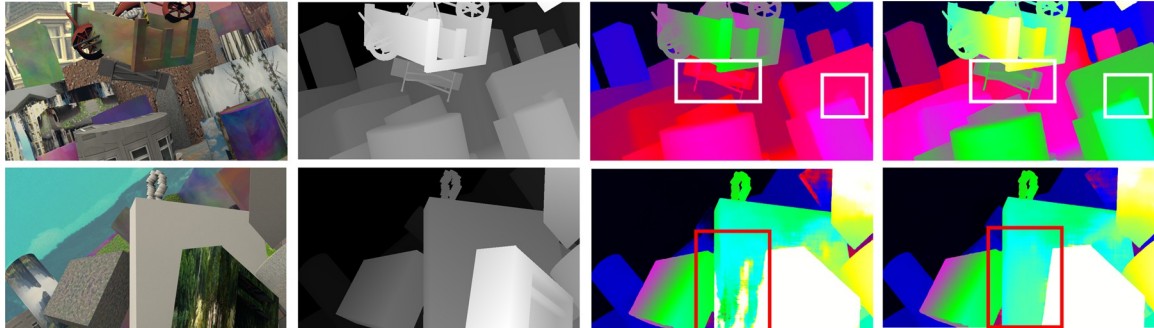

**Fig 1. Comparisons with GWC-Net [13] on Scene flow dataset.** First column: left input stereo image. Second column: the ground truth map corresponds to input image. Third column: predicted disparity map by GWC-Net. Last column: predicted disparity map by PA-Net. As shown in the white boxes in top row, PA-Net performance better in nearby objects under the guidance of PA module. In bottom row, our predicted disparity map performance better in recovering edge cues.

3. Our PA-Net achieves the accuracy of 0.775 end-point-error (EPE) on Scene flow dataset and 2.05% kitti-d1-all error on KITTI 2015 dataset, which outperforms other methods by 12%.

# Related work

## 2.1. Traditional methods

In non-end-to-end depth stereo matching algorithms, each step of traditional stereo matching can be replaced by a neural network. Some researchers have mainly focused on the use $CNN_S$ to accurately calculate the matching cost function and use the semi-global matching [14,15] method to optimize the predicted disparity map. Zbontar et al. [16] proposed a network structure called stereo matching by training a convolutional neural network (MC-CNN) to compare image patches to calculate the cost of matching by utilizing a pair of 9×9 patches. Traditional algorithms play an important role in stereo matching tasks. However, traditional algorithms generally face the problems of slow calculation speed and low matching accuracy, which greatly limits the application of stereo matching algorithm.

## 2.2. Learning-based methods

In 2015, Long *et al.* [17] achieved very good results in semantic segmentation using a fully convolutional network (FCN). Mayer *et al.* [8], inspired by the FCN, introduced an end-to-end stereo network in an optical flow prediction task. Disp-Net calculates the Euclidean distance construction loss for each pixel between the estimated disparity map and real disparity value. Cascade residual learning: A two-stage convolutional neural network for stereo matching (CRL) [18], and learning for disparity estimation through feature constancy. (iRes-Net) [19], utilized the idea of DispNetC [8] with stack refinement structures to optimize stereo results. Kendal *et al.* [20] proposed an end-to-end network GC-Net, which considers the use of context and scene geometry information in stereo matching. GC-Net is the first to concatenate the left $f_l$ and the right feature $f_r$ to form a 4D cost volume:

$$C_{concatenation}(x, y, d) = Concat\{f_l(x, y), f_r(x - d, y)\}. \tag{1}$$

Meanwhile, GC-Net transforms the stereo matching problem into a regression problem and directly realizes a refined output without post-processing. Encouraged by GC-Net, Chang *et al.* [21] proposed a PSM-Net, combining the spatial pyramid pooling and stacked 3D

hourglass structures in a stereo refinement network. In the current work, GWC-Net [13] proposes a group-wise correlation to assemble the cost volume, whose idea is splitting the features into groups and computing correlation maps group by group. The group-wise correlation is computed as

$$C_{group-wise}(d, x, y, g) = \frac{1}{N_c/N_g} \langle f_l^g(x, y), f_r^g(x - d, y) \rangle, \tag{2}$$

where $N_c$ denotes the channels of unary features and it eventually divided into $N_g$ groups along channel dimension. $\langle \cdot, \cdot \rangle$ is the internal product at all disparity levels $d$. Although the performance on several benchmarks is significantly promoted, there remains some drawbacks, including the predicted edge contour of the disparity map is not accurate enough and adopting the strategy of global attention, which is insensitive to the detailed texture information.

### 2.3. Learning-based attention methods

In recent years, researchers are encouraged by the mechanism of human attention and attempt to design some network attention architectures with a CNN to enhance the performance of feature extraction. Hu *et al.* [22] introduced a squeeze-and-excitation block to fully utilize the channel information in the network. In addition to channel attention, cbam: convolutional block attention module [23] introduced a spatial attention block to demonstrate that spatial features are vital in the network. Wang *et al.* [24] introduced a PASSR-Net to integrate super-resolution information from a stereo image pair, and proposed a PAM module in the article of PASMnet [25] to calculate the consistency score of left and corresponding right graphs along the epipolar line, and it was leveraged by many subsequent methods such as [19–26]. On the basis of PAM, Wang *et al.* [27] introduced a symmetric bi-directional parallax attention module (biPAM) to obtain cross-view information. Ying *et al.* [28] proposed a generic stereo attention module (SAM) which aims to solve the information incorporation problem. Chen *et al.* [29] addressed the stereo images with large disparity and multi types of epipolar lines issues by utilizing a cross parallax attention module (CPAM). However, the parallax information provided by binocular images has not been fully utilized in those methods. PA-Net is the first to emphasize that by improving feature expression in near-range regions is helpful to disparity prediction task.

### 2.4. Edge detection methods

Edge cues can be easily captured by human eyes to find stereo correspondences. Accurate edge contours can help discriminating between different objects or regions. Based on this observation, some works had made some progress in predicting image edge cues as a single task. Xie *et al.* [30,31] first designed an end-to-end ED network based on a VGG-16 network. Recently, Song *et al.* [32,33] combined an ED branch with stereo-matching network. However, these methods regard disparity prediction and edge detection as a multi-task learning project, those works did not establish an effective mechanism to integrate the information learned by multi-task project. As a result, the features learned by multi-task project are not effectively expressed and utilized. Focus on this problem, we construct a multi-featured integration cost volume to combine parallax features and edge features.

## Methods

As shown in Fig 2, we proposed a PA stereo matching network (PA-Net), which extends GWC-Net [13] with a PA module, edge detection branch, and multi-featured integration cost volume.

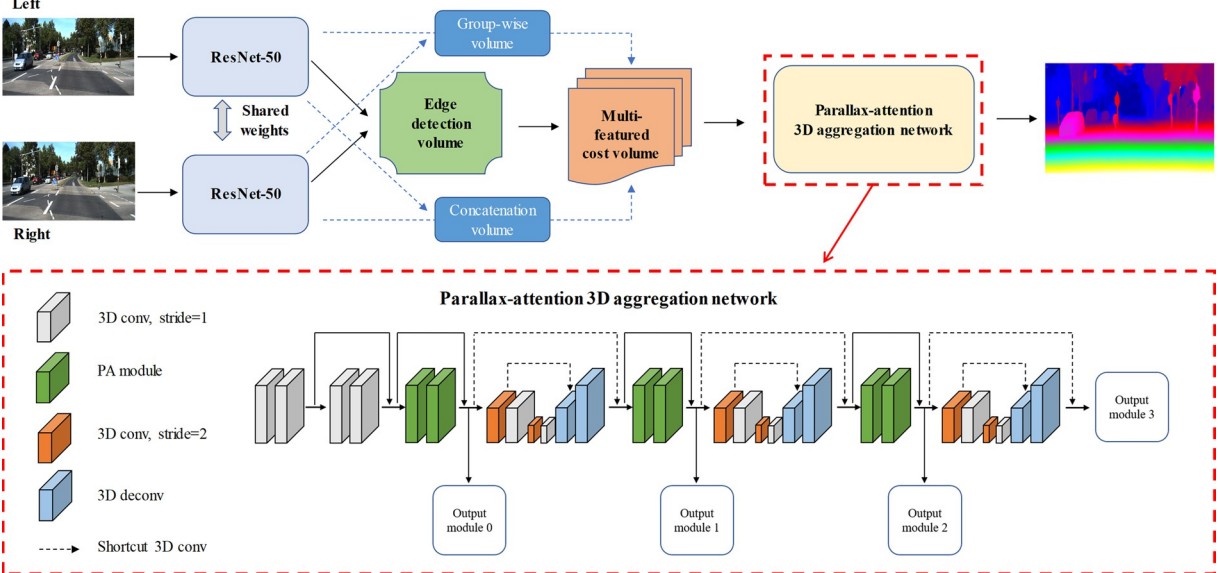

**Fig 2. Diagrammatic sketch of PA-Net.** It is constructed based on GWC-Net by adding edge-detection branch in feature extraction structure and applying PA module in the architecture of the 3D aggregation network. The left and corresponding right images are fed to a weight-sharing feature extraction pipeline, which consisting of a ResNet-50 network for feature maps calculation. It includes three branches (edge detection, group-wise correlation [13], and concatenation branches). Thereafter, a multi-featured integration cost volume is constructed by those branches and it will finally be fed into a parallax-attention 3D aggregation network for disparity regression.

## 3.1. Network architecture

The pipeline of our introduced PA network is shown in the upper half of Fig 2, it includes four parts: unary feature extraction pipeline, multi-featured integration cost volume structure, parallax-attention 3D aggregation network, and disparity prediction module. The multi-featured integration cost volume structure consists of three parts: concatenation [20], group-wise correlation [13], and edge detection volumes (details in Section 3.3). The results of the multi-featured integration cost volume are then concatenated as the input of the parallax-attention 3D aggregation network, and it will be described in Section 3.2.

The parallax-attention 3D aggregation network aims to aggregate variable disparity values, which consist of two parts: a pre-hourglass module and three parallax-attention 3D aggregation networks. The pre-hourglass module consists of two components: the primary half consists of four 3D convolutional layers with batch normalization and the ReLU [26] function, where the second part consists of two PA modules.

## 3.2. Parallax attention module

Discriminant characteristic representations are essential for understanding the scenes. However, previous studies only focus on the two-dimensional (2D) contextual information, but ignore the significance of 3D disparity features. To emphasize the value of regions with a large parallax, we introduce a PA module that encodes the disparity information to different weights, thus enhancing their illustration capability.

3D convolution layer is widely used in stereo matching tasks and it consists of 4 parts: channel dimension, disparity dimension, height and width. However, 3D filters learned within a local field that lacks contextual information in the output feature map $U$.

Based on these observations, as is shown in Fig 3, given a feature map $f \in R^{C \times D \times H \times W}$, we first conduct two transformations $f_s:f \to f_s \in R^{C \times 1 \times H \times W}, f_m:f \to f_m \in R^{C \times 1 \times H \times W}$, which represent the

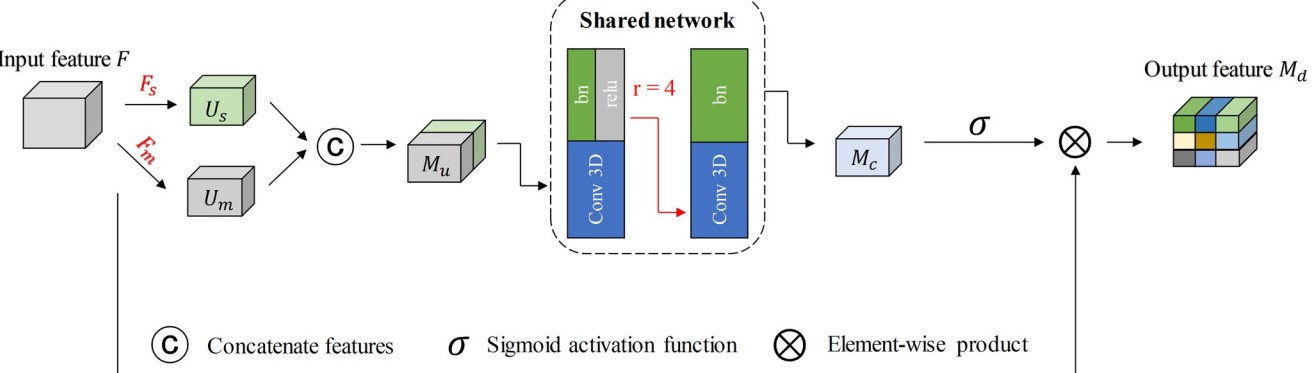

**Fig 3. Diagram of our PA module.** Given the feature *f*, PA generates parallax weights by fusing average and maximum values in disparity dimension.

importance of different position vectors. Term $f_s$ implies to select the maximum value element in disparity dimension, whereas $f_m$ denotes to calculate the mean value in disparity dimension, with regard to the $c^{th}$ channel, the value of $(i,j)$ position is calculated by:

$$f_m(c, i, j) = \frac{1}{d_{\max}} \sum_{d=1}^{d_{\max}} U_c(i^d, j^d), \qquad (3)$$

where $d_{\max}$ denotes the maximum parallax value. Thereafter, the two feature maps are concatenated in the channel dimension to obtain a mixed feature map $M_u \in R^{2C \times 1 \times H \times W}$. The mixed feature map $M_u$ can be treated as a collector of the local disparity texture information, and its function is to describe the entire parallax image.

Subsequently, the feature map is sent to a shared network, which is composed of a multi-layer perceptron with two 3×3×3 convolutional layers and it accompanies the batch normalization and ReLU [26] function. To reduce the parameter overhead, the characteristics of the middle layer are set to $\mathbb{R}^{C/r \times 1 \times H \times W}$, where *r* denotes the reduction ratio, and we set it to 4. Additionally, the disparity feature map is applied to a sigmoid function. Finally, we merge the output feature vectors with the input feature *f* using an element-wise product to obtain the final PA feature map $M_d \in R^{C \times D \times H \times W}$, which can be simplified as follows:

$$M_{dX_i} = \frac{X_i}{1 + \exp^{-(MLP\langle f_s(X_i), f_m(X_i)\rangle)}}, \qquad (4)$$

where $M_{d_i}$ denotes the value of the final $i^{th}$ position, $\langle \cdot, \cdot \rangle$ means concatenating the inner channels, and $X_i$ denotes the value of the input feature.

In comparison with traditional 3D convolutional layers, our contributions can be summarized as follows:

1. In the case of acquiring an identical receptive field, our module generates considerably fewer parameters (reduced by 25%) and consumes much less memory; consequently, the inference time of our module is faster.

2. As summarized in Table 1, our PA structure can effectively decrease the performance of EPE with a small increase of computational complexity.

3. Our PA module does not change the number of channels and the size of input features, which can be added directly to 3D convolution layers.

**Table 1. Ablation study results of PSM-Net, GWC-Net and PA-Net on the datasets of Scene flow [8].** The results of PSM-Net [21] and GWC-Net [13] are trained with published code with our batch size, evaluation settings for fair comparison.

| Model | Edge detection | PA module | EPE (px) | >1px (%) | >2px (%) | >3px (%) | Time (ms) |
|---|---|---|---|---|---|---|---|
| PSM-Net | | | 0.988 | 10.62 | 6.31 | 5.02 | 246.1 |
| | ✓ | | 0.955 | 10.27 | 6.10 | 4.85 | 251.5 |
| | | ✓ | 0.892 | 10.16 | 6.31 | 4.80 | 259.4 |
| GWC-Net | | | 0.878 | 9.25 | 5.57 | 4.35 | 210.7 |
| | ✓ | | 0.856 | 9.08 | 5.47 | 4.27 | 215.6 |
| | | ✓ | 0.792 | 8.59 | **5.24** | 4.08 | 224.4 |
| PA-Net(ours) | ✓ | ✓ | **0.775** | **8.49** | 5.26 | **3.84** | 222.4 |

## 3.3. Edge detection and multi-featured cost volume

State-of-the-art disparity estimation method works well on ordinary and clear texture regions. The matching clues in these regions are clear and can be easily captured through the context pyramid. However, as shown in Fig 1, the edge details are lost. Hence, we design an edge detection branch to help modify disparity map.

Our edge detection (ED) architecture includes three branches (group-wise, concatenation, and edge detection branches), sharing the same weights of the ResNet-50 backbone, listed in Table 1. There are four outputs in the ResNet-50 layer, for each output branch, we design a new structure that includes a 3×3 convolutional layer and 1×1 convolutional layer with batch normalization and the ReLU [26] function (we set the number of the final channel to 1 in each branch); In order to fuse the features contained in different branches, all the feature maps are concatenated to construct an edge cost volume. Finally, group-wise, concatenation, and ED features are fused to form a multi-featured integration cost volume.

As shown in Fig 4, based on the architecture of the GWC-Net [13], we added an ED branch to construct the edge volume. In contrast to the concatenation volume, within which the left and corresponding right feature maps are concatenated at unique disparity levels to form a cost volume, the ED volume is constructed by computing the similarities of the left and right

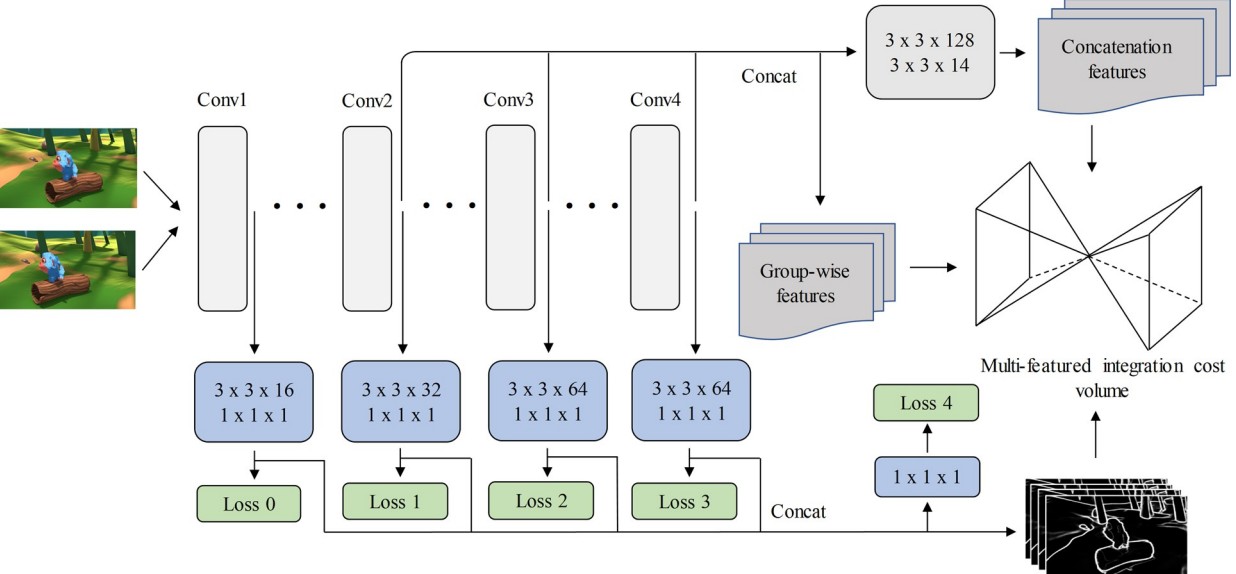

**Fig 4. Pipeline of feature extraction network.**

feature maps. For each pair of edge feature maps, the edge correlation is calculated as follows:

$$C_{edge}(d, x_i, y_j) = \frac{1}{N_c} \sum_{i>0, j>0} \langle f_{l_{edge}}(x_i, y_j), f_{r_{edge}}(x_i - d, y_j) \rangle, \tag{5}$$

where $\langle \cdot, \cdot \rangle$ denotes the internal product, $N_c$ the quantity of total channels, and $f_{l_{edge}}, f_{r_{edge}}$ the left and corresponding right feature maps, respectively. Finally, by combining all cost volumes, the multi-scale cost volume is determined as follows:

$$C_{Multi-scale} = Concat(C_{edge}, \ C_{concatenation}, \ C_{group-wise}), \tag{6}$$

where $Concat(\cdot, \cdot)$ denotes the concatenation of feature maps in the channel dimension. $C_{concatenation}$ and $C_{group-wise}$ are calculated as introduced in the Eqs of (1) and (2).

### 3.4. Output module and loss function

Summarizing all outputs of our network, it includes 4 disparity predicted maps $d_s$, to fully utilize the output feature maps, we assign different weights to each output. We first employ two convolutional layers with 3×3 and 1×1 to obtain a 1-channel output; thereafter, the output feature map is upsampled using bilinear interpolation. Finally, a softmax function is designed to calculate the disparity prediction map. Generally, the disparity smooth loss can be calculated as follows:

$$L_d(d_s, \tilde{d}_s) = \sum_{i=1}^{4} \lambda_i \left[ \frac{1}{N} \sum_{j=1}^{N} smooth_{L1}(d_s^j - \tilde{d}_s^j) \right], \tag{7}$$

where $\lambda_i$ denotes the weight for the $i^{th}$ output disparity prediction map, N represents the total number of pixels in one image, and $d_s^j$ is the $j^{th}$ element with ground truth $\tilde{d}_s^j$. The $smooth_{L1}$ loss is computed as follows:

$$Smooth_{L_1}(x) = \begin{cases} \dfrac{x^2}{2}, \ if \ |x| < 1 \\ |x| - 0.5, \ otherwise \end{cases} \tag{8}$$

Since the information of object edge contour in images is conducive to the parallax prediction task, we propose an edge detection loss for end-to-end learning:

$$L_e(x) = \frac{1}{N} \sum_{j=1}^{N} \begin{cases} \log(1 - P(x^j)), \ if \ y^j = 0 \\ \log P(x^j), \qquad otherwise \end{cases} \tag{9}$$

where $x^j$ and $y^j$ represent the activation value and ground truth edge probability at pixel j, respectively. $P(x)$ is the standard sigmoid function, and N denotes the total number of pixels in one image. Fusing the edge feature information extracted from different output layers, our edge loss function can be formulated as:

$$L_e = \sum_{k=1}^{3} \beta_k L_e(x^k) + L_e(x^{fuse}), \tag{10}$$

where $x^k$ is the activation value from stage $k$ while $x^{fuse}$ denotes the last edge output. $\beta_k$ is the weight of stage k (equals to 0.2, 0.4, and 0.6 here). Since we are working under a disparity prediction task setup, we want to fuse the edge detection loss and disparity prediction loss together. Therefore, we design a double hierarchical loss weighing scheme, the total loss is

calculated as:

$$L = \gamma_0 L_d + \gamma_1 L_e, \tag{11}$$

with $\gamma_0$ is the weight of total disparity prediction loss and it set to 1, $\gamma_1$ denotes the total edge detection loss weighted 0.1.

## Experiments

In this section, we evaluated our PA-Net with distinctive settings on the Scene Flow [8] and the KITTI datasets [9,10]. Sections 4.1 and 4.2 show the experimental setup of proposed network on the KITTI and Scene Flow datasets. In Section 4.3, we set up a series of ablation experiments using different methods to test the performance of our PA module. In Section 4.4, we add our edge detection volume to PSM-Net and GWC-Net to validate the importance of the multi-featured integration cost volume.

### 4.1. Experimental setup

We implemented our architectures using the PyTorch tools. All methods were trained using Adam ($\beta_1 = 0.9$, $\beta_2 = 0.99$). Owing to the limitation of experimental conditions, the batch size of our network was set to 4, and we optimized all the models with two Nvidia RTX 2080ti GPUs using $256 \times 512$ random crops from the input image pair. The utmost disparity value was set to 192, whereas the coefficients of the four outputs were set to $\lambda_1 = 0.5$, $\lambda_2 = 0.5$, $\lambda_3 = 0.7$, and $\lambda_4 = 1.0$. We tend to set the model on the Scene Flow dataset for a total of 16 epochs in which the learning rate was 0.001 and downscaled by 2 when the number of epochs 10, 12, and 14. For KITTI [9,10] dataset, the pre-trained model on Scene Flow [8] datasets was utilized to train another 300 epochs. The preliminary learning rate was 0.001, it is down-scaled by 10 when exceeding 200 epochs.

### 4.2. Dataset

*1) Scene Flow [8]:* A dataset series of artificial stereo datasets. There are three subsets in the dataset: Driving, Flyingthings3D, and Monkaa, containing 35454 images for training and 4370 images for testing with Height = 540 and Width = 960, along with ground truth maps. The trained network model has a strong generalization ability because the number of pictures in the Scene Flow dataset is sufficiently large. The results of visualization and comparisons for Scene Flow [8] are as shown in Fig 1.

2) *KITTI 2012 and KITTI 2015 [9,10]:* Real-word driving scene dataset using Lidar scanning to obtain the three-dimensional coordinates of space points. KITTI 2012 includes 194 training stereo correspondences and 195 testing pairs. KITTI 2015 comprises 200 stereo correspondences for training and testing. The training dataset is divided into two parts, the first section consists of 180 pairs for training and the relaxation groups of images for validation. More than that, we made the corresponding edge detection label dataset for end-to-end edge detection task. The results of the visualization and comparisons are shown in Fig 5, and we submit the results predicted by PA model on the validation set of the KITTI official website. The comparison results for the test set are summarized in Tables 2 and 3. It shows that our PA-Net achieves better results than PSMNet [21], GwcNet [13] and PASMNet [25].

### 4.3 Ablation study for parallax attention module

In this section, to validate the performance of the PA module, we evaluated the PA module with different stereo matching strategies. Moreover, we set a series of ablation experiments to

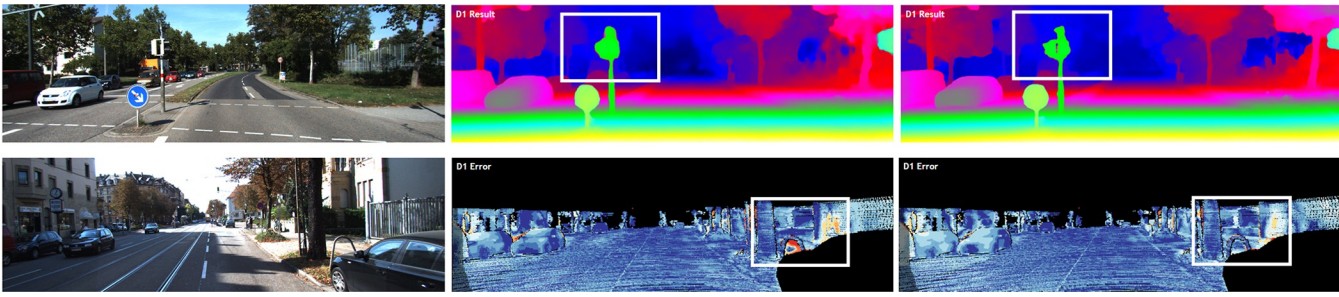

**Fig 5. Results visualization and comparisons on KITTI 2015 [10].** Left column: the input left image. Middle column: predicted color disparity and error map by GWC-Net. Right column: predicted color disparity and error map by PA-Net. As shown in the white boxes, PA-Net performance better than previous work in areas with large parallax such as the iron railing in front of the car.

explore the best settings for the number of PA modules. [13,21] were selected as reference models by adding PA modules.

PA module can be directly used in 3D convolution layer, since our model will not change the number of channels and image size. The experimental results demonstrate that, on the premise of a small increase in calculation time, our PA-Net performs better than previous works. As summarized in Table 1, we select the classic methods [13,21] as the conference models which include two variables (edge detection structure and PA module). Meanwhile, the prediction result of an EPE on the Scene Flow dataset is decreased by 9.71% in the model of [21] and decreased by 11.0% in the model of [13] after adding the PA modules. Fig 6, depicts the training and validation curves of PA-Net, GWC-Net and PSM-Net on KITTI 2015 dataset from epoch 100 to 300. We can easily observe that the loss curve of PA-Net decreases more smoothly than previous works and produce consistent gains in performance which are sustained throughout the training process. Moreover, we see that PA-Net performs better than those works when the networks are trained to start fitting, and it will last achieve a highest accuracy.

To select an optimal value of PA modules to configure the networks. As shown in Table 4, which indicates the consequence of PA-Net with different numbers of PA modules. When the number of PA modules is larger than 6, the increase in accuracy becomes minor. Considering the amount of calculation and memory consumption, we selected half a dozen PA modules as the ultimate structure.

**Table 2. Performance comparison of KITTI 2012 [9] test set.** The GWC-Net [13] and PSM-Net [21] are trained with the same batch size as our method for fair comparison.

| Methods | >2px (%) | | >3px (%) | | >5px (%) | | Mean Error (px) | | Time (s) |
|---|---|---|---|---|---|---|---|---|---|
| | Noc | All | Noc | All | Noc | All | Noc | All | |
| DispNetC [8] | 7.38 | 8.11 | 4.11 | 4.65 | 2.05 | 2.39 | 0.9 | 1.0 | 0.06 |
| MC-CNN-acrt [16] | 3.90 | 5.45 | 2.43 | 3.63 | 1.64 | 2.39 | 0.7 | 0.9 | 67 |
| GC-Net [20] | 2.71 | 3.46 | 1.77 | 2.30 | 1.12 | 1.46 | 0.6 | 0.7 | 0.9 |
| iResNet [19] | 2.69 | 3.34 | 1.71 | 2.16 | 1.06 | 1.32 | 0.5 | 0.6 | 0.12 |
| PSMNet [21] | 2.68 | 3.20 | 1.68 | 2.09 | 1.05 | 1.21 | 0.5 | 0.5 | 0.6 |
| GWCNet [13] | 2.21 | 2.88 | 1.40 | 1.81 | 0.85 | 1.11 | 0.5 | 0.5 | 0.32 |
| Edge-stereo [32] | 2.32 | 2.88 | 1.46 | 1.83 | 0.83 | **1.04** | **0.4** | 0.5 | 0.32 |
| PA-Net | **2.20** | **2.82** | **1.36** | **1.78** | **0.80** | 1.09 | 0.5 | **0.5** | 0.33 |

**Table 3. Performance comparison of KITTI 2015 [10] test set.** The GWC-Net [13] and PSM-Net [21] are trained with the same batch size as our method for fair comparison.

| Methods | All (%) | | | Noc (%) | | | Time(s) |
|---|---|---|---|---|---|---|---|
| | D1-bg | D1-fg | D1-all | D1-bg | D1-fg | D1-all | |
| DispNetC [8] | 4.32 | 4.41 | 4.34 | 4.11 | 3.72 | 4.05 | 0.06 |
| GC-Net [20] | 2.21 | 6.16 | 2.87 | 2.02 | 5.58 | 2.61 | 0.9 |
| CRL [18] | 2.48 | 3.59 | 2.67 | 2.32 | 3.12 | 2.45 | 0.47 |
| iResNet [19] | 2.14 | 3.45 | 2.36 | 1.94 | 3.20 | 2.15 | 0.22 |
| PSMNet [21] | 1.98 | 4.87 | 2.32 | 1.71 | 4.51 | 2.24 | 0.41 |
| GWCNet [13] | 1.85 | 4.14 | 2.23 | 1.71 | 3.75 | 2.05 | 0.32 |
| PASMNet [25] | 2.04 | 4.33 | 2.41 | 1.88 | 3.91 | 2.22 | 0.5 |
| Edge-stereo [32] | 1.84 | **3.30** | 2.08 | 1.69 | **2.94** | 1.89 | 0.32 |
| PA-Net | **1.73** | 4.05 | **2.05** | **1.63** | 3.59 | **1.86** | 0.33 |

## 4.4 Ablation study for multi-featured cost volume

In this section, we apply several critical modifications to the feature extraction network compared to [13]. Specifically, we design a multi-featured integration cost volume structure that consists of three parts: ED cost, group-wise cost, and concatenation cost volumes.

The experimental results in Table 1 demonstrate that by adding our edge detection structure, the parameters of EPE loss can be reduced appropriately. As summarized in Table 1, the prediction results of EPE on the Scene Flow dataset are decreased by 3.34% in the model of [21] and decreased by 2.50% in the model of [13] after adding the ED modules.

Based on [13], we can conclude from several experiments that if we set the channel number of the group-wise volume as 32, we can obtain an exceptional performance. The experimental consequences in Table 5 demonstrate that the EPE is decreased by adding the correct channels of the concat volume. The best EPE is 0.574 (concat channels are *14×2*) in the dataset of KITTI 2015 and 0.616 (concat channels are *16×2*) in KITTI 2012. Considering both the accuracy of disparity prediction and computer memory usage, we selected *14×2* as the ultimate channel of the concatenation volume.

## 4.5 Analysis and interpretation

While PA blocks have been empirically shown to improve network performance, we would like to provide an explain how the parallax attention mechanism operates in practice. To provide a clearer picture of the behavior of PA blocks, in this section we apply several examples from GWC-Net model and examine the different distributions of sensitive respective region between 3D convolutional layers and PA blocks. We then exhibit their distribution maps in Fig 7, which is trained in the dataset of KITTI 2015.

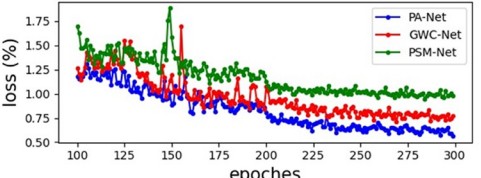 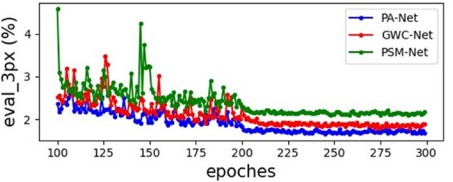

**Fig 6. Training curves of PA-Net, GWC-Net and PSM-Net on KITTI 2015 dataset from epoch 100 to 300.** PA-Net exhibits improved optimization characteristics and more stabled in training.

**Table 4. Results of EPE by adding different number of PA modules on Scene flow dataset.**

| Model | The numbers of PA moduels | Scene Flow EPE (px) | Time (s) |
|---|---|---|---|
| PA-Net | 0 | 0.856 | 0.329 |
| | 1 | 0.849 | 0.331 |
| | 4 | 0.816 | 0.334 |
| | 6 | **0.784** | 0.336 |
| | 10 | 0.790 | 0.340 |
| | 16 | 0.801 | 0.346 |

We make the following observations about how the parallax attention mechanism works in 3D feature extraction stage. First, traditional 3D convolutional layer is used to adopting global receptive field mechanism, which will guide the network to pay fair attention to different features. However, in the practice life, we can easily draw a conclusion that objects closer to us will have a greater impact. As shown in Fig 7, compared with trees and houses far away, people and cars in near region should be paid more attention. But we can observe a phenomenon that the lighted regions in 3D feature map have covered every corner of the image, which is not in line with objective reality. Second, for PA blocks, we redistribute the values in the parallax dimension to make full use of context information. For the region with large parallax value, the proportion of its value will be larger after redistribution. In the third line of Fig 7, lighted regions are concentrated in areas with large parallax such as roads, cyclist and cars, indicating that the network pays more attention to these areas. PA blocks successfully focus on objects with large parallax through the weight redistribution strategy in parallax dimension.

## Discussion

Intuitively, our method not only improves the accuracy of disparity prediction globally, we also ahcieve the following advantages: Firstly, in the case of acquiring an identical receptive field with traditional 3D convolutional layer, our module generates considerably fewer parameters (reduced by 25%) and consumes much less memory. More than that, PA module can be easily utilized in other works because it will not change the size of the feature image. Secondly, as shown in Fig 1, our structure efficiently improves the accuracy of disparity prediction in near-range regions by improving 3D feature expression. Lastly, in order to making full use of the edge information learned by two-dimensional feature extraction network, we propose a novel edge detection branch and multi-featured integration cost volume. It is demonstrated that based on our model, edge detection project is conducive to improve the accuracy of disparity estimation. As Table 2 shows, compared with GWC-Net, our method performs better in two-pixel error, three-pixel error, and five-pixel error on the KITTI 2012 dataset. Compared with PSM-Net, the disparity map's percentage of outliers averaged over all ground truth pixels (D1-all) is reduced by 11.6%, and the running speed is increased by 19.5%.

**Table 5. Ablation study results of PA-Net with different settings on the dataset of KITTI.**

| Model | Concatenation channels | Kitti2015 EPE (%) | Kitti2012 EPE (%) | Time (ms) |
|---|---|---|---|---|
| PA-Net (edge = 4, group = 32) | 10×2 | 0.612 | 0.633 | 215.7 |
| | 12×2 | 0.586 | 0.621 | 218.5 |
| | 14×2 | **0.574** | 0.617 | 222.4 |
| | 16×2 | 0.578 | **0.616** | 225.9 |
| | 18×2 | 0.575 | 0.619 | 229.4 |

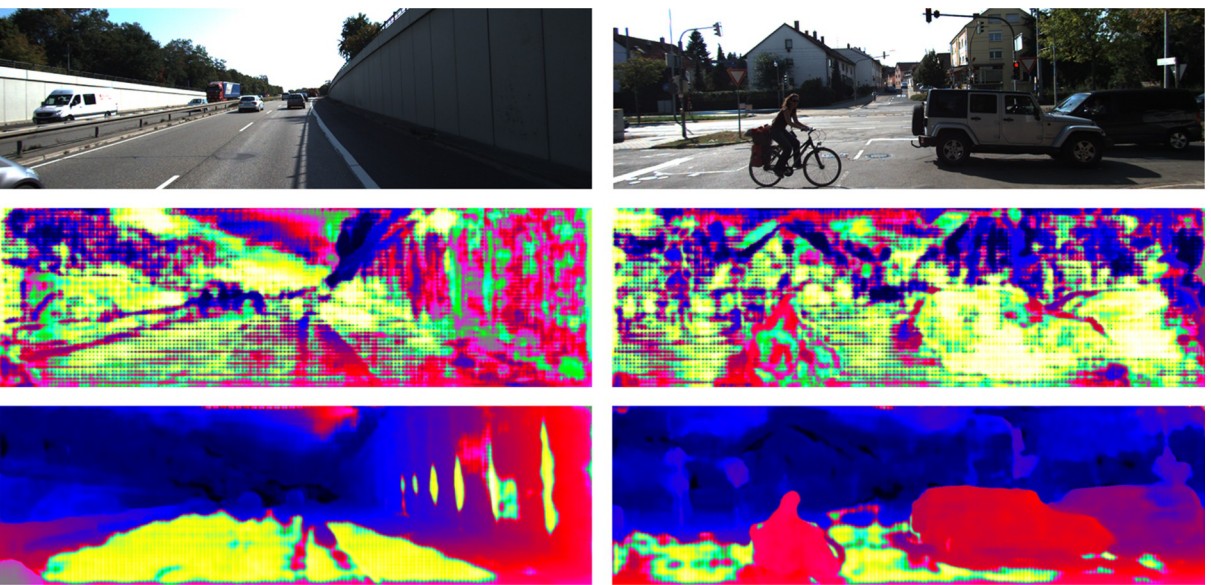

**Fig 7. Attention distribution map in KITTI 2015.** First line: the input left image. Middle line: attention distribution feature map extracted by 3D convolutional layers. Third line: attention distribution feature map extracted by PA blocks.

## Conclusions

In this paper, we proposed a high-precision and practical stereo matching network, PA-Net, for end-to-end disparity prediction. Our net emphasizes that by improving feature expression in near-range regions is helpful to disparity prediction task. PA-Net performs better than previous networks by utilizing the edge detection layer, PA module, and multi-featured cost volume. It is demonstrated that based on our model, edge detection task is conducive to improve the accuracy of disparity estimation task. PA-Net achieves better accuracy than GWC-Net on the Scene Flow and KITTI datasets.

## Acknowledgments

We would like to thank Editage (*www.editage.com*) for English language editing.

## Author Contributions

**Conceptualization:** Xuefei Yu.

**Data curation:** Xuefei Yu.

**Formal analysis:** Xuefei Yu.

**Funding acquisition:** Xuefei Yu.

**Investigation:** Xuefei Yu.

**Methodology:** Xuefei Yu.

**Project administration:** Xuefei Yu.

**Resources:** Xuefei Yu, Zedong Huang.

**Software:** Xuefei Yu.

**Supervision:** Xuefei Yu, Jinan Gu, Zedong Huang, Zhijie Zhang.

**Validation:** Xuefei Yu.

**Visualization:** Xuefei Yu.

**Writing – original draft:** Xuefei Yu.

**Writing – review & editing:** Xuefei Yu.

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
