## [Decision Letter · Decision Letter 0]

14 Jul 2021

PONE-D-21-20417

Parallax attention stereo matching network based on the improved group-wise correlation stereo network

PLOS ONE

Dear Dr. jinan,

Thank you for submitting your manuscript to PLOS ONE. After careful consideration, we feel that it has merit but does not fully meet PLOS ONE’s publication criteria as it currently stands. Therefore, we invite you to submit a revised version of the manuscript that addresses the points raised during the review process.

We look forward to receiving your revised manuscript.

Kind regards,

Jyotismita Chaki, PhD

Academic Editor

PLOS ONE

Journal Requirements:

"This research was funded by the National Natural Science Foundation of China (No.51875266)"

5. Please ensure that you refer to Figure 6 in your text as, if accepted, production will need this reference to link the reader to the figure.

6. We note that Figures 1,4, and 6 in your submission contain copyrighted images. All PLOS content is published under the Creative Commons Attribution License (CC BY 4.0), which means that the manuscript, images, and Supporting Information files will be freely available online, and any third party is permitted to access, download, copy, distribute, and use these materials in any way, even commercially, with proper attribution. For more information, see our copyright guidelines: http://journals.plos.org/plosone/s/licenses-and-copyright.

a. You may seek permission from the original copyright holder of Figures 1,4, and 6 to publish the content specifically under the CC BY 4.0 license. 

Additional Editor Comments:

We have received comments on your paper. Reviewers' comments are appended below. Your manuscript requires revision. If you undertake the revision and resubmit the revised manuscript, I would be pleased to review your revised manuscript. Please submit a list of changes or a rebuttal against each point raised by the reviewers along with the revised manuscript.

Reviewers' comments:

Reviewer's Responses to Questions

**Comments to the Author**

1. Is the manuscript technically sound, and do the data support the conclusions?

Reviewer #1: Partly

Reviewer #2: Yes

2. Has the statistical analysis been performed appropriately and rigorously? 

Reviewer #1: Yes

Reviewer #2: Yes

3. Have the authors made all data underlying the findings in their manuscript fully available?

Reviewer #1: Yes

Reviewer #2: Yes

4. Is the manuscript presented in an intelligible fashion and written in standard English?

Reviewer #1: Yes

Reviewer #2: Yes

5. Review Comments to the Author

Reviewer #1: In this paper, the authors proposed a parallax attention network for stereo matching. They also proposed an edge detection module and a multi-scale cost volume technique to further improve the accuracy. Experimental results validate the effectiveness of the proposed modules. The proposed method achieves better performance than PSMnet and GwcNet.

Major Concerns:

1) Parallax attention has been widely used for various of stereo image processing tasks, such as stereo image super-resolution [c1-c3], stereo matching [c4], stereo image dehazing [c5], and stereo color transfer [c6]. The authors are suggested to provide a comprehensive review of the recent progress of parallax attention in the related work section.

[c1] Symmetric parallax attention for stereo image super-resolution, CVPRW 2021.

[c2] A stereo attention module for stereo image super-resolution, SPL 2020.

[c3] Feedback Network for Mutually Boosted Stereo Image Super-Resolution and Disparity Estimation, arxiv 2021.

[c4] Parallax Attention for Unsupervised stereo correspondence learning, TPAMI 2020.

[c5] BidNet: Binocular image dehazing without explicit disparity estimation, CVPR 2020.

[c6] Asymmetric stereo color transfer, ICME 2021.

2) Since parallax attention has already been used for stereo matching task in [c4], the main difference between the proposed network and PASMnet [c4] should be well addressed and discussed.

Minor Concerns:

1) In table 1, the resolution of the 2nd to 4th branches of the PA module is 1/4 x 1 x 1/4H x 1/4 W x 32(or64). The first 1/4 is confusing. Is it the batch dimension?

Reviewer #2: This paper proposes a novel parallax attention method for binocular disparity estimation. The effectiveness of the proposed method is validated through extensive experiments. Comparative results also show that the proposed PA-Net is superior than PSMnet and GWCNet. I recommend an acceptance to this paper upon some minor revisions.

1) It should be noted that parallax attention mechanism has already been used for disparity estimation in PASMnet [r1]. However, the method proposed in this paper is significantly different from that in [r1]. The authors are strongly suggested to discuss the difference between their method and PASMnet.

2) Since this paper focuses on the parallax attention, some recent advances in parallax attention should be reviewed in the related work (e.g., symmetric parallax attention [r2], cross parallax attention [r3], stereo attention [r4] and so on) to make the survey of the literature more comprehensive.

3) The layout of the tables and figures in this paper need to be further improved. A table cannot be placed on two separate pages.

Refs:

[r1] Parallax attention for unsupervised stereo correspondence learning, TPAMI 2020.

[r2] Symmetric parallax attention for stereo image super-resolution, arxiv 2020.

[r3] Cross parallax attention for stereo image super-resolution, TMM 2020.

[r4] A stereo attention module for stereo image super-resolution, SPL 2020.

6. PLOS authors have the option to publish the peer review history of their article (what does this mean?). If published, this will include your full peer review and any attached files.

Reviewer #1: No

Reviewer #2: No

---

## [Author Response · Author response to Decision Letter 0]

19 Jul 2021

Reviewer #1: We are very grateful to your comments for the manuscript, your suggestion is of great help to our work. According to your advice, we amended the relevant part in manuscript. We have taken the following measures to this advice: Firstly, we have changed the expression of “Despite the excellent performance of the attention module, it rarely appears in stereo matching network” in the beginning of our previous abstract to “Despite the excellent performance of the attention module, the parallax information provided by binocular images has not been fully utilized.” Secondly, we have made a comprehensive review of the recent progress of parallax attention in the introduction and related work section. We introduced the progress of recent works such as PASMnet, PASSRnet, SISRnet, etc. For more details, please check the latest manuscript. Thanks again for your advices.

Reviewer #2 We are very grateful to your comments for the manuscript, and we have made a comprehensive review of the recent progress of parallax attention in the introduction and related work section. We introduced the progress of recent works such as PASMnet, PASSRnet, SISRnet, etc. For more details, please check the latest manuscript. Thanks again for your advices.

---

## [Decision Letter · Decision Letter 1]

16 Sep 2021

PONE-D-21-20417R1

Parallax attention stereo matching network based on the improved group-wise correlation stereo network

PLOS ONE

Dear Dr. jinan,

Thank you for submitting your manuscript to PLOS ONE. After careful consideration, we feel that it has merit but does not fully meet PLOS ONE’s publication criteria as it currently stands. Therefore, we invite you to submit a revised version of the manuscript that addresses the points raised during the review process.

We look forward to receiving your revised manuscript.

Kind regards,

Jyotismita Chaki, PhD

Academic Editor

PLOS ONE

Journal Requirements:

1. We suggest you thoroughly copyedit your manuscript for language usage, spelling, and grammar. If you do not know anyone who can help you do this, you may wish to consider employing a professional scientific editing service. 

Additional Editor Comments (if provided):

Based on the reviewer comments I am requesting you to submit the revised manuscript.

**Review Comments to the Author**

Reviewer #1: All my concerns have been addressed by the authors. I recommend an acceptance without further revisions.

Reviewer #3: This paper presents a new network for the stereo correspondence problem. From the comparison, it does look like the proposed framework outperforms other works. The structure of the network is clearly explained. Generally, the work looks good. However, the writing of the paper is far from ready.

First of all, the paper jump into the specific technique and problem directly from the beginning. There is not much background introduction and concept explanation for normal readers. If the reader (even doing computer vision research) are not familiar with this specific research, it will be very difficult to understand the work and follow the writing. I do suggest the author to use the introduction section to clearly explain the problem, instead of review some papers.

Secondly, this paper made the same mistake as lots of machine learning papers. It only explains the proposed network structure then jump directly into the experiment, using the result to show the effectiveness of the new network. There is no clear explanation why this network is designed in this specific way. It is pretty much like, I try this and “luckily” the result is good. For a good academic writing, there should be a clear motivation and discussion on the design. All the decision should be clearly justified.

Some other comments on the details:

1. In the abstract, the paper targets at “ill-posed areas (weak texture, repeat texture and occlusion regions)”. However in the paper, all those problems are not mentioned specifically in the experiment.

2. In the abstract, how the “novel edge detection branch and multi-scale cost volume” will “obtain finer texture features”. Is there a justification for this claim or “common sense”?

3. The English needs to be polished. For example, I am not sure the following expression are correct or not, please check: “word goal characteristic”, “aggressive universal overall performance”, “conference models”, “competitive advantage”. And there are lots of long sentences which should be broken into separate ones (for example line 124-127). And in line 217, “Which” should be merged into previous sentence. Line 309, “Table 5, demonstrate”, no comma. There are lots of small mistakes.

4. Line 77, “Our PA-Net outperformed the state-of-the-art GWC-Net [11] in ill-posed regions”, this should be put into the contribution.

5. Line 45-54 list quite a few papers, however, the logic behind this discussion is not clear. How these literature review helps the overall argument.

6. Line 135-136 should have an overall introduction about the whole methodology.

7. Table 2 needs more detailed discussion before concluding into the listed benefit.

8. Line 257-259, what do you mean “challenging regions”. Line 265, Figure 6 should be Figure 5.

9. Line 292, Figure 5 should be Figure 6

10. Line 305 and Table2 should be consistent, model [10] or [2]?

11. The reference format should be consistent. The last few doesn’t have page no, authors’ name are also not in the right format. Please check carefully.

Reviewer #4: 1. Line 73. Achieving competitive results on the dataset is not a major contribution.

2. Figure 1 is too far away from where is first referred. The proposed architecture is quite simple, it’s not clear what is the proposed novelty. The figure refers to the PA-Net abbreviation which can’t be found inside the figure.

3. Please extend the argumentation in the design chose of the architecture.

4. Table 1 is extremely long. I could have been replaced by a better design of Figures 1, 2, 3.

5. May general conclusion is that the manuscript is well prepared and there are a few small ideas that are proposed. However, the overall results are just competitive, i.e. quite similar with the state-of-the-art. Therefore, I would prefer to see an extended discussion on why such a small gain is achieved. Why would someone chose the proposed method and not other state-of-the-art method. What are the main advantages of employing PA-NET?

Reviewer #5: The primary contribution of the paper is an attention module that takes into account some of the available 3D disparity information. It builds on the work of several authors, most heavy on Kendall et al. 2017 who introduced a “Cost Volume” feature description that attempted to create a 3D volume from stereo information by concatenating features from the left image with features from the right image at some image location shifted by the depth. This produced a Depth x Height x Width x Features tensor.

Here authors add an attention network that uses this “Cost Volume” to drive attention, but not directly. They first collapse the depth dimension to two values, the mean and max. They show that this features is an improvement on previous methods on the KITTI dataset and also show its general applicability to two other datasets.

I have not been able to fully understand all of the details, especially the edge detection feature. I do not feel confident that a reader would be able to reproduce this part of the work from the details given. However the analysis performed is logical a fairly thorough, the results appear sound and could support the authors conclusions if the methodological issues were cleared up (also with the important caveats below). Therefore I strongly believe that this paper is not ready for publication but believe it can be ready with some minor but important modifications.

Major issues. The paper is very hard to follow (not unusual for a deep learning paper with so many details to the implementation). The main issue seems to be that the authors are not using the terminology in the papers they cite. The authors may have a good reason for this but it would be helpful if they either revised their paper with matching terminology or altered the reader to the difference.

“concatenation cost, group-wise correlation [11], and our proposed edge 149 detection volumes (details in Section 3.3).”

What is the “concatenation cost” or group-wise correlation? The paper cited only contains a “Cost Volume”. Section 3.3 implies that the concatenation cost is equivalent to the “Cost Volume”. I am still in the dark as to what the group-wise correlation refers to.

As said in the introduction I could not follow the edge detection work. Why is the edge branch called an edge branch? Did you run an edge detector on the images before passing the images into the network? As written it looks like you ran two slightly different networks in parallel.

Adding an edge detection module appears to degrade the performance of the PA module (Table 2) can you comment on this?

Comments on the main claims (Abstract)

“Particular, we advocate for a parallax attention module in three dimensional (disparity, height and width) level that aims to aggregate variable disparity values. “

As I understand it your system does not in fact use the entire depth range (or a single disparity value) but two values derived from this information.

“Meanwhile, finding correct correspondences in ill-posed areas (weak texture, repeat texture, and occlusion regions) remains an arduous task. “

How does your system address this problem? All your results are based on errors across an entire image not selected areas that are ill-posed.

“To reinforce the precision of the disparity prediction in these challenging areas, in the present study, we propose a parallax attention stereo matching algorithm based on the improved group-wise correlation stereo network to learn the disparity content from a stereo correspondence. “

You don’t propose a new group-wise correlation, you use an pre-existing group-wise correlation method and extend it with an attention mechanism.

You repeatedly use the term “multi-scale” to describe your algorithm. I have always understood the term multi-scale to mean combining images of different sizes (at least in this context). What you a developing appears more “multi-featured” that “multi-scale”.

Minor issues

“In recent years, researchers have attempted to combine the human attention mechanism with a 112 CNN to enhance the performance of feature extraction”

The actual human attention mechanism is still not understood, an is probably very different the systems in machine learning. Can you alter this to say something like “human-inspired”

Update citations in tables (e.g. table 2, 3)

Table 2: 1027 should be 10.27 ?

---

## [Author Response · Author response to Decision Letter 1]

14 Oct 2021

Thank you for giving us the opportunity to submit a revised draft of the manuscript “Parallax attention stereo matching network based on the improved group-wise correlation stereo network” for publication in the Journal of “PLOS ONE”. We appreciate the time and effort that you and the reviewers dedicated to providing feedback on our manuscript and are grateful for the insightful comments and valuable improvements to our paper. We have studied comments carefully and have made correction which we hope meet with approval. Revised portion are marked in red in the paper.

---

## [Decision Letter · Decision Letter 2]

12 Nov 2021

PONE-D-21-20417R2Parallax attention stereo matching network based on the improved group-wise correlation stereo networkPLOS ONE

Dear Dr. jinan,

Thank you for submitting your manuscript to PLOS ONE. After careful consideration, we feel that it has merit but does not fully meet PLOS ONE’s publication criteria as it currently stands. Therefore, we invite you to submit a revised version of the manuscript that addresses the points raised during the review process.

We look forward to receiving your revised manuscript.

Kind regards,

Jyotismita Chaki, PhD

Academic Editor

PLOS ONE

Journal Requirements:

Reviewers' comments:

Reviewer's Responses to Questions

**Comments to the Author**

1. If the authors have adequately addressed your comments raised in a previous round of review and you feel that this manuscript is now acceptable for publication, you may indicate that here to bypass the “Comments to the Author” section, enter your conflict of interest statement in the “Confidential to Editor” section, and submit your "Accept" recommendation.

Reviewer #4: All comments have been addressed

2. Is the manuscript technically sound, and do the data support the conclusions?

Reviewer #3: Yes

Reviewer #4: Yes

3. Has the statistical analysis been performed appropriately and rigorously? 

Reviewer #3: Yes

Reviewer #4: Yes

4. Have the authors made all data underlying the findings in their manuscript fully available?

Reviewer #3: Yes

Reviewer #4: Yes

5. Is the manuscript presented in an intelligible fashion and written in standard English?

Reviewer #3: No

Reviewer #4: No

6. Review Comments to the Author

Reviewer #3: Thanks for the authors making the revision following our previous comments. Some of the problems are very well addressed. However, there are still lots of problems in the writing, not only typo or grammar, but also the logic behind the writing. I really recommend the authors to get some help. Some examples are list here:

1. The first part of the abstract should address the background, research problem and motivation. However the logic that the authors follow is that parallax information and edge contour information are not fully used. This shouldn’t be the research problem. The target is to improve the precision of disparity prediction. The author should discuss what is the current status of accuracy, how bad it is. The parallax and edge information is the methodology that the authors choose to exploit. For any research, it shouldn’t target at just experimenting some methods, that only give you the route for solving the problem not the problem itself. There are other parts in the abstract and introduction section also follow this logic which should be revised.

2. In the introduction section, “However, complicated manual production steps limit their improvement.” This statement concludes too quick. It needs more details (discussion) to justify why and where these methods are not good. Probably one more sentence will be enough. Also for “and “Secondly, adopting the strategy of global attention, without increasing attention to important areas.” Why this is not good, also missing a part in the logic of the discussion.

3. In section 2.4, the sentence “Observation that edge detection project is conducive to improve the accuracy of disparity estimation.” is not linked clearly with the adjacent part. Also, the last sentence in the same paragraph, the author meant to criticise the Song’s work, it only mentioned multi-task instead of integration, how this become a problem? One more sentence could conclude this statement, however it is missing. For this literature review section, discussion is very important, and it should have a clear logical link, end to end, landing on the problems that this paper trying to solve.

4. Section 3.3, “where matching cues are clear can be easily captured through the context pyramid.”

5. Section 3.3, Figure 4 caption, “Pipeline of feature extraction network. Which includes three branches (edge detection, group-wise, and concatenation feature branches).”

6. Figure 2, “Edge detection voume”, typo problem.

It is impossible for me to give a comprehensive list of problems in the writing. The authors should carefully check every detail of the draft. Certainly the research work is good. It worthy taking time to polish the writing to achieve the level of publication in this journal.

Reviewer #4: The authors answered all my comments, however, I would recommend to further polish the text and add the corresponding punctuation after the equations.

In the future, I would recommend to the authors to answer each reviewer’s comment, not just to mention that the manuscript was modified somehow, e.g.: “We have made the latest changes in the method part.”. Please add an answer to the comment and clearly present what was modified in the manuscript, not just refer to the manuscript!

7. PLOS authors have the option to publish the peer review history of their article (what does this mean?). If published, this will include your full peer review and any attached files.

Reviewer #3: No

Reviewer #4: **Yes: **Ionut Schiopu

---

## [Author Response · Author response to Decision Letter 2]

15 Nov 2021

Thank you for giving us the opportunity to submit a revised draft of the manuscript “Parallax attention stereo matching network based on the improved group-wise correlation stereo network” for publication in the Journal of “PLOS ONE”. We appreciate the time and effort that you and the reviewers dedicated to providing feedback on our manuscript and are grateful for the insightful comments and valuable improvements to our paper. We have studied comments carefully and have made correction which we hope meet with approval. Revised portion are marked in red in the paper.

---

## [Decision Letter · Decision Letter 3]

5 Jan 2022

PONE-D-21-20417R3Parallax attention stereo matching network based on the improved group-wise correlation stereo networkPLOS ONE

Dear Dr. jinan,

Thank you for submitting your manuscript to PLOS ONE. After careful consideration, we feel that it has merit but does not fully meet PLOS ONE’s publication criteria as it currently stands. Therefore, we invite you to submit a revised version of the manuscript that addresses the points raised during the review process.

We look forward to receiving your revised manuscript.

Kind regards,

Jyotismita Chaki, PhD

Academic Editor

PLOS ONE

Reviewers' comments:

Reviewer's Responses to Questions

**Comments to the Author**

1. If the authors have adequately addressed your comments raised in a previous round of review and you feel that this manuscript is now acceptable for publication, you may indicate that here to bypass the “Comments to the Author” section, enter your conflict of interest statement in the “Confidential to Editor” section, and submit your "Accept" recommendation.

Reviewer #6: (No Response)

2. Is the manuscript technically sound, and do the data support the conclusions?

Reviewer #6: Partly

3. Has the statistical analysis been performed appropriately and rigorously? 

Reviewer #6: Yes

4. Have the authors made all data underlying the findings in their manuscript fully available?

Reviewer #6: Yes

5. Is the manuscript presented in an intelligible fashion and written in standard English?

Reviewer #6: No

6. Review Comments to the Author

Reviewer #6: The paper proposes a stereo matching framework that incorporates a disparity prediction model, the edge detection layer, PA module, and multi-featured cost volume. The authors made strong emphasis on improving feature expression in near-range regions to help disparity prediction tasks.

Main Review:

Strengths:

The proposed method improves the accuracy of disparity estimation task.

Weakness:

-Major concerns:

1. The only difference between the proposed method and GWC-Net is the PA module, but is it absolutely necessary? According to the qualitative results shown in the main text, it is hard to interpret why the proposed PA module is effective which only brings marginal improvements.

2. The visual comparison is not enough to support the superiority of the proposed method.

3. The technical contributions of this paper are unclear. In the last few paragraphs, I think the logical relationship is not clearly explained, which leads to confusion when reading. I would like to see a description about why it works, as well as the innovations and differences from other related methods.

-Minor concerns:

1. visualization of the results. All the visualizations are relatively low resolution, which is confusing and makes it hard to compare across methods and against GT.

2. In the paper, ‘our theory’ and ‘in this study’ appeared several times, please replace them with ‘out method’, ‘in this paper’

3. In the summarization of contributions, for the third contribution, revise it to: Our PA-Net achieves the accuracy of ()% on Scene flow dataset and ()% on KITTI 2015 dataset, which outperforms other methods by ()%.

4. The paper also has some minor issues in writing, in the section of Introduction

-whose goal is to -> which aims at computing

-“Learning-based stereo-matching methods through exploring feature representations and aggregation algorithms for matching costs”, this sentence is not complete.

- due to the limited learning features-> due to limited learning capability

- the most easily recognized and utilized-> the most easily recognized

-Avoid using their

-However, their methods regard….as a multi-task learning project, in this way …. effective mechanism to fuse them-> However, these methods regard disparity prediction and edge detection as a multi-task learning project. Yet, features learned in such multi-task pipelines cannot be fully exploited, which poses a great need for an effective fusion mechanism.

-In the autonomous driving task-> In the context of autonomous driving,

-Thus, it requires the disparity estimation model to provide more attention to this region.->To address this problem, more attention should be assigned to this kind of region in the disparity estimation model.

-In this study->In this paper

-we demonstrate that by designing a high-quality and efficient module for stereo matching…->we propose a high-quality and efficient module for stereo matching and our method achieves better performance on SceneFlow and KITTI than previous methods.

-It is demonstrated that based on our parallax attention stereo matching network->It is demonstrated that our parallax attention stereo matching

-edge detection task is conducive to improve the accuracy of disparity estimation? Please make the sentence more understandable.

5. Section 3.2

-“However, each of the learned 3D filters with a local field that the output feature map U is unable to learn contextual information.” The whole sentence is not clear, please reorganize it. If I am understanding right. How about revising as the following: However, 3D filters learned within a local field that lacks contextual information in the output feature map U.

-when in the c-th chanel->with regard to the c-th channel

-We can treat M_u as a collector of..->The mixed feature map M_u can be treated as a collector of the local disparity texture information, and its function is to describe the entire parallax image

-“Subsequently, the descriptors” You mentioned before as “a feature map”, which is inconsistent with ‘descriptors’ used here. Please avoid confusion like this in the whole manuscript.

Also, “Subsequently” occurred twice in a paragraph. Replace the second one with ”additionally”.

-Our benefits can be summarized as follows-> our contribution can be summarized as follows\\

-shorter->faster

-“increase the performance of end-point-error” increase? Is the higher value of end-point-error, the better?

-Our proposed PA module-> Our PA module

-So we can add it directly to 3D convolution layers->, which can be added directly to 3D convolution layers

6. Section 3.3

-ED features were fused-> ED features are fused

-Please elaborate the equation of group-wise cost volume after equation 6.

7. Section 4.1

-It decreased 10 times->it is down-scaled by 10 when exceeding 200 epochs

-In the section of discussion, please revise “our method performance” to “our method performs...”

-Table 2 caption, “..are trained in our batch-size”->..are trained with the same batch size as our method. Please check through the whole manuscript.

-Table 3 caption, citation[10] should follow the word “test”

8. Section 4.3

- “the PA module techniques”, is PA module a technique？

- because->since

- Our proposed PA-Net performance is better-> our PA-Net performs better

- To select the proper number of->To select an optimal value of..

9. Section 4.4

- A collection of->several

Summary Of The Review:

Based on theses observation, I believe this paper is slightly below the the acceptance threshold. And I suggest a re-submission to the journal of PLOS-ONE.

7. PLOS authors have the option to publish the peer review history of their article (what does this mean?). If published, this will include your full peer review and any attached files.

Reviewer #6: No

---

## [Author Response · Author response to Decision Letter 3]

12 Jan 2022

We are very grateful to your comments for the manuscript. To prove the necessity of PA module, we add a new section (Section 4.5 Analysis and Interpretation) to exhibit PA blocks performs better than previous works. Moreover, we add a new figure map (Fig 7. Attention distribution map in KITTI 2015.) which provides an explain how the parallax attention mechanism operates in practice and why it is effective.

---

## [Decision Letter · Decision Letter 4]

26 Jan 2022

Parallax attention stereo matching network based on the improved group-wise correlation stereo network

PONE-D-21-20417R4

Dear Dr. jinan,

We’re pleased to inform you that your manuscript has been judged scientifically suitable for publication and will be formally accepted for publication once it meets all outstanding technical requirements.

Kind regards,

Jyotismita Chaki, PhD

Academic Editor

PLOS ONE

Additional Editor Comments (optional):

I am happy to inform you that reviewers are satisfied with the revised manuscript. Thus the manuscript is provisionally accepted for publication.

Reviewers' comments:

Reviewer's Responses to Questions

**Comments to the Author**

1. If the authors have adequately addressed your comments raised in a previous round of review and you feel that this manuscript is now acceptable for publication, you may indicate that here to bypass the “Comments to the Author” section, enter your conflict of interest statement in the “Confidential to Editor” section, and submit your "Accept" recommendation.

Reviewer #6: All comments have been addressed

2. Is the manuscript technically sound, and do the data support the conclusions?

Reviewer #6: Yes

3. Has the statistical analysis been performed appropriately and rigorously? 

Reviewer #6: Yes

4. Have the authors made all data underlying the findings in their manuscript fully available?

Reviewer #6: Yes

5. Is the manuscript presented in an intelligible fashion and written in standard English?

Reviewer #6: Yes

6. Review Comments to the Author

Reviewer #6: The text deserves to be proofread again to correct typos before publishing.

Here are some typos in the re-submitted manuscript:

1: In section 4.3, "we can easily observation.."  we can easily observe..

2: Fig 6 caption, "PA-Net exhibit" PA-Net exhibits

A careful reading and grammar check would definitely increase the quality of the paper.

7. PLOS authors have the option to publish the peer review history of their article (what does this mean?). If published, this will include your full peer review and any attached files.

Reviewer #6: No

---

## [Editor Report · Acceptance letter]

31 Jan 2022

PONE-D-21-20417R4 

Parallax attention stereo matching network based on the improved group-wise correlation stereo network 

Dear Dr. Gu:

I'm pleased to inform you that your manuscript has been deemed suitable for publication in PLOS ONE. Congratulations! Your manuscript is now with our production department. 

Kind regards, 

on behalf of

Dr. Jyotismita Chaki 

Academic Editor

PLOS ONE